

# Clinical characteristics and prognosis of basaloid squamous cell carcinoma of the lung: a population-based analysis

Guangda Yuan[1], Cheng Zhan[2], Yiwei Huang[2], Donglin Zhu[1], Hongya Xie[1], Tengteng Wei[1], Tao Lu[2], Qun Wang[2], Yong Yang[1] and Yimeng Zhu[1]

[1] Department of Thoracic Surgery, The Affiliated Suzhou Hospital of Nanjing Medical University, Suzhou,Jiangsu, China
[2] Department of Thoracic Surgery, Zhongshan Hospital, Fudan University, Shanghai, China

Corresponding authors
Yong Yang, yangy_suzhou@126.com
Yimeng Zhu,
zhuym_suzhou@126.com

## ABSTRACT

**Background.** This study analyzed the clinical features and prognosis of basaloid squamous cell carcinoma of the lung (BSC), and constructed a nomogram to predict the prognoses of patients.

**Methods.** The information of pure BSC patients was obtained in the Surveillance, Epidemiology, and End Results database between 2004 and 2015. Then, it was evaluated, and compared with the data of lung squamous cell carcinoma (SCC), lung large cell carcinoma (LCC) and lung adenocarcinoma (LAC) patients. Subsequently, we used univariate and multivariate analyses to investigate the independent factors related to the prognoses of patients with BSC and constructed a nomogram to verify the prognoses.

**Results.** A total of 425 patients diagnosed with BSC were enrolled. Compared with patients with SCC, LCC and LAC, the mean survival time of BSC patients was better than all of them. Compared with SCC, there were significant differences between the characteristics of grade ($P < 0.001$), total stage ($P < 0.001$), T stage ($P < 0.001$), N stage ($P < 0.001$), M stage ($P < 0.001$), surgery ($P < 0.001$), radiotherapy ($P < 0.001$), and chemotherapy ($P < 0.001$), while BSC also had significantly different clinical characteristics from LCC and LAC. Univariate and multivariate survival analyses showed that age ($P < 0.001$), T stage ($P < 0.001$), N stage ($P = 0.009$), M stage ($P < 0.001$), and surgery ($P < 0.001$) were independent prognostic factors of BSC. The survival of patients undergoing lobectomy was significantly better than sublobar resection, with an OR of 0.389 (0.263–0.578). We constructed a nomogram with a C-index of 0.750 (95% confidence interval) based on the results of multivariate analysis. The calibration curves based on nomogram scores indicated that the nomogram could accurately predict the prognosis of patients.

**Conclusions.** BSC had unique clinical and prognostic features. T stage, N stage, M stage, age, and surgery were independently associated with overall survival (OS). Lobectomy was a relative ideal choice for patients with BSC. The nomogram effectively predicted the OS at 1-, 3-, and 5-years.

## INTRODUCTION

Lung cancer is the most commonly diagnosed cancer worldwide, as well as the leading cause of cancer deaths. Worldwide in 2018, it accounted for 2.1 million new cases and 1.8 million deaths. Squamous cell carcinoma is one of the major well-studied histological subtypes of lung cancer (*Allemani et al., 2018*; *Bray et al., 2018*). However, basaloid squamous cell carcinoma of the lung (BSC), as a rare subtype of lung squamous cell carcinoma, is less studied, and the clinical features and prognostic factors remain unclear.

BSC is an uncommon histological variant of lung cancer composed of cells exhibiting cytological and tissue architectural features of both squamous cell lung carcinoma and basal cell carcinoma, while the proportion of squamous cell components is less than 50% (*Brambilla et al., 2001*; *Wang et al., 2011*; *Crapanzano et al., 2011*). It is reported that BSC accounts for 3.9%–5.2% of all lung squamous carcinomas and has unique clinical characteristics such as a high rate of metastasis and death, according to previous researches. Apart from this, BSC was once described with overlap features of large cell carcinoma (LCC) (*Travis et al., 2015*; *Vignaud, 2016*). Up to this date, no relevant literatures have reported the differences of clinical features with LCC and other non-small cell lung cancer (NSCLC). In this study, we compared clinicopathological characteristics of related lung cancer subtypes in detail, then we used univariate and Cox hazards regression analyses to identify risk factors affecting overall survival (OS) of BSC. We further developed a nomogram of patients with BSC based on the results of survival analysis to better predict the prognoses of patients.

## MATERIALS AND METHODS

### Ethic statement

We obtained permission to use data files from the public database of the Surveillance, Epidemiology, and End Results (SEER) database. Thus, our research was exempted by the Ethics Committee of Suzhou Municipal Hospital.

### Data extraction

Data of all primary pure basaloid squamous cell carcinoma patients (ICD-O-3: 8083/3) between 2004 to 2015 were identified by the SEER*Stat software (v8.3.5, https://seer.cancer.gov/seerstat/) from the SEER database (http://seer.cancer.gov/). Exclusion criteria were: (1) pathological types of non-pure-type basaloid squamous cell carcinoma; (2) unknown aspects regarding differentiation, stage, and treatment methods; and (3) a history of tumors in other sites (Fig. 1).

We extracted and analyzed the data regarding patients' race, sex, age, grade, TMN stage, surgery type, radiotherapy, and chemotherapy. The total stage, T stage, N stage, and M stage of all patients were manually restaged according to the 8th edition of the American Joint Committee on Cancer (AJCC) lung cancer staging project.

Chi-square tests were used for comparison of multi-class variables like race between basaloid squamous cell carcinoma and other types of lung carcinomas. Rank sum tests were used for comparing two categorical variables or ordered variables. The quantitative variables

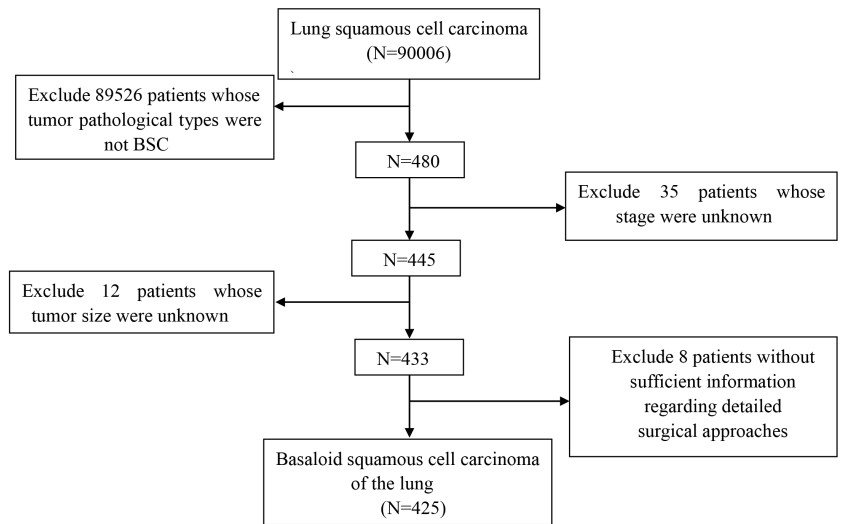

**Figure 1** **The flow diagram of the selection process for the study cohort.**

of age were compared by the variance analysis method. In the analysis of prognostic factors for BSC, we used Kaplan–Meier analyses and log-rank tests for univariate analyses, and Cox model tests for multivariate analyses. The above analyses were performed using SPSS (version 25) software (SPSS, Chicago, IL, USA), all of which were two-sided tests, while $P < 0.05$ was considered to be statistically significant. R language (*R Core Team, 2018*) was used to generate and validate the nomogram, while the main packages used were rms and Hmisc (*Sun et al., 2017*; *Wang et al., 2018*).

# RESULTS

## Comparison of clinical features between BSC and other types of NSCLC

After screening, we enrolled 425 patients with BSC of pure type and 90006, 6997 and 160638 patients with SCC, LCC and LAC, respectively. Survival curves indicated that the survival of BSC patients was significantly better than those of SCC, LCC and LACpatients (Fig. 2 and Fig. S1). As shown in Table 1, 257 males and 168 females were included in the BSC group, with a median age of 70.15 years and interquartile range (IQR) of 59.87–80.43 years. In total, 358 (84.2%) were white. The grades included 0.5% grade I (well differentiated), 10.8% grade II (moderately differentiated), 64.5% grade III (poorly differentiated), and 3.0% grade IV (undifferentiated). Most BSC patients (42.1%) were AJCC stage I, 14.8% were stage II, 23.8% were stage III, and 19.3% were stage IV. BSC patients had significantly less well differentiated tumors ($P < 0.001$), less N+ disease ($P < 0.001$), fewer distant metastases ($P < 0.001$), lower proportion of radiotherapy ($P < 0.001$) and chemotherapy ($P < 0.001$), but a higher percentage of radical surgical resection ($P < 0.001$) than those of SCC and LAC patients. Conversely, only LCC patients had more undifferentiated tumors ($P < 0.001$), while much lower proportion of surgery ($P < 0.001$), radiotherapy ($P < 0.001$) and chemotherapy ($P < 0.001$) (Table 1).

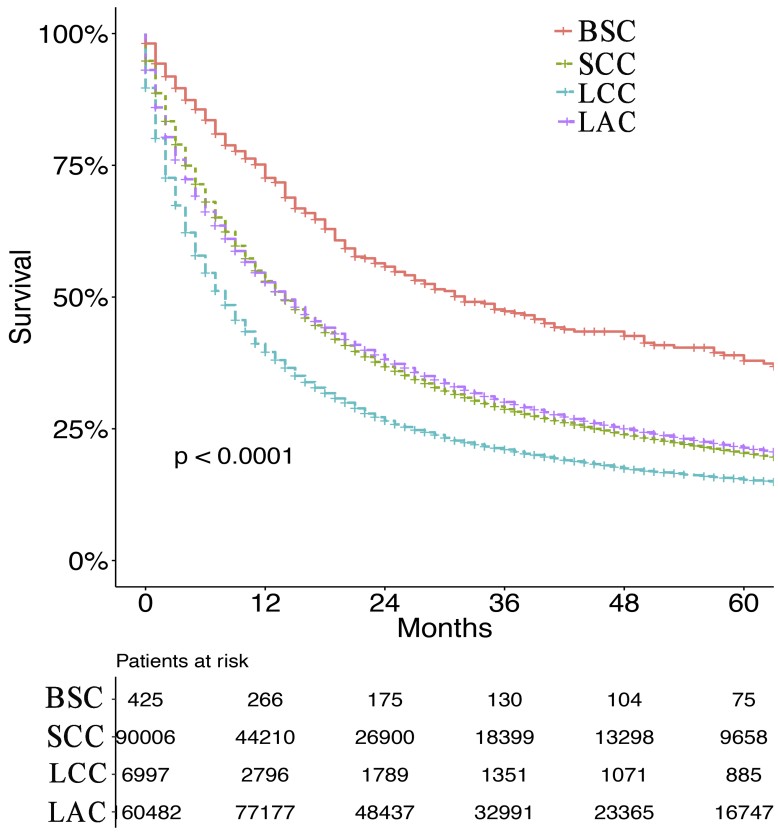

**Figure 2** **Survival for BSC, SCC, LCC and LAC.** The survival curves of basaloid squamous cell carcinoma of the lung (BSC), squamous cell carcinoma (SCC), large cell carcinoma (LCC) and lung adenocarcinoma (LAC).

## Analyses of BSC prognostic factors

We used univariate analyses to investigate possible prognostic factors in patients with BSC. As shown in Table 1, there was a statistically significant correlation between age ($P < 0.001$), grade ($P < 0.001$), total stage ($P < 0.001$), T stage ($P < 0.001$), N stage ($P < 0.001$), M stage ($P < 0.001$), surgery ($P < 0.001$), radiotherapy ($P < 0.001$) and chemotherapy ($P = 0.013$) with prognoses of BSC. In other words, elder age, lower differentiation, and a higher total stage meant the worse prognosis. In addition, as shown in Fig. 3A, univariate analyses showed that patients who underwent surgery had better prognoses than patients who did not, and were similar to patients receiving other surgical treatments ($P < 0.001$). Particularly, the patients with Stage I to Stage IV who underwent lobectomy had better benefits than those undergoing sublobar resection (Fig. 3B), including for total stage ($P < 0.0001$), Stage I ($P = 0.00016$), Stage II ($P = 0.0012$), Stage III, and IV (both $P = 0.03$) (Figs. 3C, 3D and 3E). Sex ($P = 0.257$) and race ($P = 0.077$) for the prognosis of BSC were not statistically significant (Table 2).

The data revealed that the factors of age ($P < 0.001$), T stage ($P < 0.001$), N stage ($P = 0.009$), M stage ($P < 0.001$), and surgery ($P < 0.001$) with statistical significance

**Table 1** Comparison of the clinipathological characteristics of basaloid squamous cell carcinoma (BSC), squamous cell carcinoma of the lung (SCC), large cell carcinoma of the lung (LCC) and lung adenocarcinoma (LAC).

| Characteristics | BSC | SCC | p value | LCC | p value | LAC | p value |
|---|---|---|---|---|---|---|---|
| **Race, n (%)** | | | 0.351 | | 0.039 | | 0.049 |
| White | 358 (84.2) | 75,272 (83.6) | | 5,682 (81.2) | | 128,685 (80.1) | |
| Black | 43 (10.1) | 10,617 (11.8) | | 999 (14.3) | | 18,644 (11.6) | |
| Other | 24 (5.7) | 4,117 (4.6) | | 316 (4.5) | | 13,309 (8.3) | |
| **Age, median [IQR]** | 70.15 (59.87–80.43) | 70.41 (60.66–80.16) | 0.139 | 67.92 (56.87–78.97) | 0.303 | 68.37 (57.27–79.47) | 0.376 |
| **Sex, n (%)** | | | 0.503 | | 0.443 | | <0.001 |
| Male | 257 (60.5) | 55,849 (62.1) | | 4,099 (58.6) | | 78,527 (48.9) | |
| Female | 168 (39.5) | 34,157 (37.9) | | 2,898 (41.4) | | 82,111 (51.1) | |
| **Grade, n (%)** | | | <0.001 | | <0.001 | | <0.001 |
| Well differetiated | 2 (0.5) | 1,974 (2.2) | | 15 (0.2) | | 11,955 (7.4) | |
| Moderately differetiated | 46 (10.8) | 26,606 (29.6) | | 80 (1.1) | | 36,005 (22.4) | |
| Poorly differetiated | 274 (64.5) | 32,307 (35.9) | | 2,115 (30.2) | | 43,291 (27.0) | |
| Undifferetiated | 13 (3.0) | 648 (0.7) | | 2,299 (32.9) | | 968 (0.6) | |
| Unknown | 90 (21.2) | 28,471 (31.6) | | 2,488 (35.6) | | 68,419 (42.6) | |
| **Total stage, n (%)** | | | <0.001 | | <0.001 | | <0.001 |
| I | 179 (42.1) | 27,683 (30.8) | | 1,460 (20.9) | | 41,408 (25.8) | |
| II | 63 (14.8) | 10,946 (12.2) | | 534 (7.6) | | 10,560 (6.6) | |
| III | 101 (23.8) | 26,297 (29.2) | | 1,858 (26.6) | | 32,528 (20.2) | |
| IV | 82 (19.3) | 25,080 (27.8) | | 3,145 (44.9) | | 76,142 (47.4) | |
| **T stage, n (%)** | | | <0.001 | | <0.001 | | 0.001 |
| T1 | 133 (31.3) | 9,151 (10.2) | | 1,292 (10.2) | | 43,558 (27.1) | |
| T2 | 139 (32.7) | 66,275 (73.6) | | 2,346 (73.6) | | 47,776 (29.8) | |
| T3 | 68 (16.0) | 3,596 (4.0) | | 880 (4.0) | | 22,072 (13.7) | |
| T4 | 85 (20.0) | 10,984 (12.2) | | 2,479 (12.2) | | 47,232 (29.4) | |
| **N stage, n (%)** | | | <0.001 | | <0.001 | | <0.001 |
| N0 | 269 (63.3) | 9,151 (10.2) | | 2,847 (40.7) | | 72,104 (44.9) | |
| N1 | 50 (11.8) | 66,275 (73.6) | | 663 (9.5) | | 14,110 (8.8) | |
| N2 | 84 (19.8) | 3,596 (4.0) | | 2,628 (37.5) | | 54,487 (33.9) | |
| N3 | 22 (5.1) | 10,984 (12.2) | | 859 (12.3) | | 19,937 (12.4) | |
| **M stage, n (%)** | | | <0.001 | | <0.001 | | <0.001 |
| M0 | 343 (80.7) | 58,972 (65.5) | | 3,854 (55.1) | | 84,496 (52.6) | |
| M1 | 82 (19.3) | 31,034 (34.5) | | 3,143 (44.9) | | 76,142 (47.4) | |
| **Surgery, n (%)** | | | <0.001 | | <0.001 | | <0.001 |
| Not performed | 144 (33.9) | 60,336 (67.0) | | 5,041 (72.0) | | 112,857 (70.3) | |
| Lobectomy | 199 (46.8) | 20,978 (23.3) | | 1,377 (19.7) | | 35,329 (22.0) | |
| Sublobar resection | 62 (14.6) | 6,153 (6.8) | | 438 (6.3) | | 10,982 (6.8) | |
| Pneumonectomy | 20 (4.7) | 2,539 (2.8) | | 141 (2.0) | | 1,470 (0.9) | |

**Table 1** (*continued*)

| Characteristics | BSC | SCC | *p* value | LCC | *p* value | LAC | *p* value |
|---|---|---|---|---|---|---|---|
| **Radiotherapy,** *n* (%) | | | <0.001 | | <0.001 | | <0.001 |
| No | 104 (24.5) | 4 (0.0) | | 3,980 (56.9) | | 98,823 (61.5) | |
| Yes | 321 (75.5) | 90,002 (100.0) | | 3,017 (43.1) | | 61,815 (38.5) | |
| **Chemotherapy,** *n* (%) | | | <0.001 | | <0.001 | | <0.001 |
| No | 134 (31.5) | 53,403 (59.3) | | 3,944 (56.4) | | 88,220 (54.9) | |
| Yes | 291 (68.5) | 36,603 (40.7) | | 3,053 (43.6) | | 72,418 (45.1) | |

**Notes.**

*P* value for chi-square test.

BSC, Basaloid squamous cell carcinoma; SCC, squamous cell carcinoma; LCC, large cell carcinoma of the lung; LAC, lung adenocarcinoma; IQR, interquartile range.

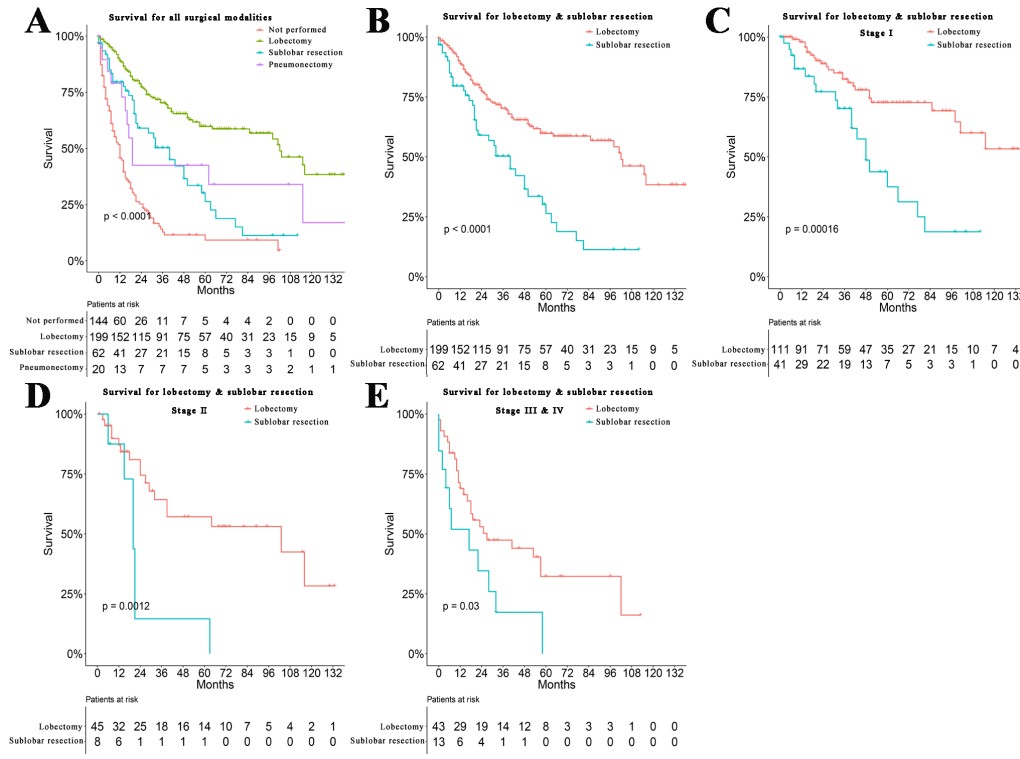

**Figure 3** **Survival for surgical modalities.** (A) Survival analyses for patients with different surgeries and without surgery. (B) Survival analyses for patients with lobectomy and sublobar resection classified by tumor histology and tumor stage. (C) Stage I (1-, 3-, 5-year survival rate of lobectomy vs. sublobar section: 82.0%, 53.2%, 31.5% vs. 70.7%, 46.3%, 17.1%). (D) Stage II (1-, 3-, 5-year survival rate of lobectomy vs. sublobar section: 71.1%, 40.0%, 31.1% vs. 75.0%, 12.5%, 12.5%). (E) Stage III and IV (1-, 3-, 5-year survival rate of lobectomy vs. sublobar section: 67.4%, 32.6%, 18.6% vs. 46.2%, 7.7%, 0.0%).

using univariate analysis were found to be independent factors according to multivariate analyses (Table 2). Multivariate analyses revealed that the older patient and the higher the TMN stage, the worse the prognosis. Excluding the significant effects of T3 ($P = 0.003$), T4 ($P < 0.001$) and N2 ($P = 0.001$), the remaining T stage and N stage of the prognoses of patients were similar. Compared with T1 and N0, the odds ratios (ORs) were T2: 1.240

**Table 2  Univariate and multivariate Cox proportional hazards analysis.**

| Patient characteristics | Univariate analysis | | Multivariate analysis | |
|---|---|---|---|---|
| | HR (95%CI) | *p* value | HR (95%CI) | *p* value |
| **Race** | | 0.077 | | – |
| White | Reference | | – | – |
| Black | 1.251 (0.811–1.931) | 0.311 | – | – |
| Other | 0.502 (0.248–1.019) | 0.056 | – | – |
| **Age** | | <0.001 | 1.032 (1.017–1.048) | <0.001 |
| **Sex** | | 0.257 | | – |
| Male | Reference | | – | – |
| Female | 0.855 (0.652–1.121) | | – | – |
| **Grade** | | <0.001 | | 0.248 |
| Well differetiated | Reference | | Reference | |
| Moderately differetiated | 0.776 (0.181–3.322) | 0.733 | 2.179 (0.404–11.756) | 0.365 |
| Poorly differetiated | 0.994 (0.245–4.033) | 0.993 | 1.894 (0.369–9.712) | 0.444 |
| Undifferetiated | 1.748 (0.375–8.139) | 0.477 | 4.198 (0.689–25.591) | 0.120 |
| Unknown | 2.291 (0.537–9.421) | 0.251 | 2.101 (0.401–10.988) | 0.379 |
| **AJCC 8th T stage** | | <0.001 | | <0.001 |
| T1 | Reference | | Reference | |
| T2 | 1.052 (0.735–1.507) | 0.780 | 1.240 (0.856–1.797) | 0.256 |
| T3 | 1.909 (1.274–2.862) | 0.002 | 1.935 (1.257–2.980) | 0.003 |
| T4 | 2.689 (1.879–3.849) | <0.001 | 2.364 (1.559–3.585) | <0.001 |
| **AJCC 8th N stage** | | <0.001 | | 0.009 |
| N0 | Reference | | Reference | |
| N1 | 1.678 (1.088–2.587) | 0.019 | 1.623 (1.018–2.586) | 0.042 |
| N2 | 3.218 (2.360–4.390) | <0.001 | 1.905 (1.300–2.792) | 0.001 |
| N3 | 5.013 (2.855–8.804) | <0.001 | 1.826 (0.951–3.505) | 0.071 |
| **AJCC 8th M stage** | | <0.001 | | <0.001 |
| M0 | Reference | | Reference | |
| M1 | 4.253 (3.170–5.704) | <0.001 | 2.399 (1.695–3.393) | <0.001 |
| **Surgery** | | <0.001 | | <0.001 |
| Not performed | Reference | | Reference | |
| Lobectomy | 0.187 (0.136–0.258) | <0.001 | 0.389 (0.263–0.578) | <0.001 |
| Sublobar resection | 0.442 (0.304–0.643) | <0.001 | 0.889 (0.567–1.394) | 0.608 |
| Pneumonectomy | 0.384 (0.209–0.706) | <0.001 | 0.614 (0.317–1.189) | 0.148 |
| **Radiotherapy** | | <0.001 | | |
| No | Reference | | Reference | |
| Yes | 2.173 (1.631–2.895) | <0.001 | 1.076 (0.768–1.508) | 0.669 |
| **Chemotherapy** | | | | |
| No | Reference | | Reference | |
| Yes | 1.423 (1.076–1.882) | 0.013 | 0.725 (0.506–1.039) | 0.080 |

**Notes.**

*$P$ value for chi-square test. $C$-index $= 0.750$.

The total stage was not an independent variable related to T, N, and M stage, therefore it was excluded in the multivariate analysis.

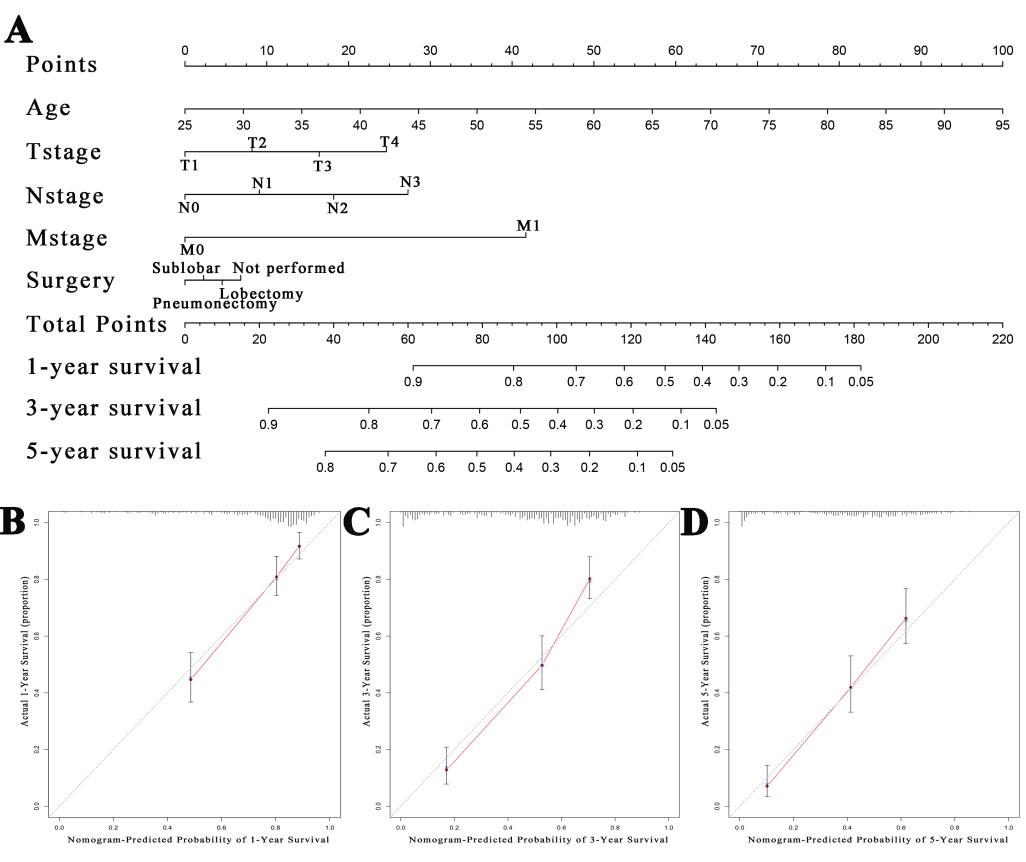

**Figure 4 Nomogram and overall survival nomogram calibration curves.** (A) Nomogram for prediction of 1-, 3- and 5- year overall survival of patients with basaloid squamous cell carcinoma of the lung (BSC). Calibration plots of the nomogram prediction of (B) 1-year, (C) 3-year and (D) 5-year overall survival nomogram calibration curves. The red line represents equality of the observed and predicted probability.

(0.856–1.797), T3: 1.935 (1.257–2.980), T4: 2.364 (1.559–3.585), and N1: 1.623 (1.018–2.586), N2: 1.905 (1.300–2.792), and N3: 1.826 (0.951–3.505), respectively. The prognoses of patients undergoing lobectomy were significantly better than other surgical styles, with an OR of 0.389 (0.263–0.578) compared with nonsurgical or other surgical patients. Unlike the results of univariate analysis, multivariate analysis showed that radiotherapy and chemotherapy were not independent prognostic factors.

## Production and inspection of the nomogram

We successfully constructed a nomogram based on the above independent predictors of patient outcomes (Fig. 4A). According to the patients' age, T stage, N stage, M stage, and surgery, we visually calculated the patient's 1- (Fig. 4B), 3- (Fig. 4C) and 5-year (Fig. 4D) survival probabilities. The $C$-index of this nomogram was 0.750 as determined by the discriminant test. The consistency test showed that the 3-year and 5-year survival rates predicted by the nomogram were in good agreement with the actual 3-year and 5-year survival rates, and the slope of the consistency curve was close to 1.

## DISCUSSION

We conducted an in-depth analysis of BSC using the patients' data from the SEER database and we found that there were significant statistical differences with SCC, LCC and LAC in terms of race, grade, total stage, T stage, N stage, M stage, surgery, radiotherapy and chemotherapy. We also found that age, T stage, N stage, M stage and surgery were independent influencing factors for the prognoses of patients with BSC. We then plotted a nomogram. Consistency detection proved that the nomogram effectively predicted the 1-, 3- and 5-year survival probabilities of patients, while the nomogram scores effectively discriminated the patients' survivals.

BSC is an invasive subtype of squamous cell carcinoma that can be detected in the proximal bronchi (*Wang et al., 2011*; *Kim et al., 2003*). Unlike other previous studies (*Brambilla et al., 1992*), we verified that the prognosis of BSC was better than SCC, LCC and LAC. In this study, we found that the prognosis of BSC in a population-based cohort was better than SCC, LCC, and LAC. However, there are some previous studies reported opposite results to ours (*Brambilla et al., 2014*; *Moro-Sibilot et al., 2008*), perhaps due to that the number of cases varied, and most of other studies focused on the patients with surgery. Meanwhile, BSC has a significant lower TNM stage than other lung cancers according to our results.

In this population-based study, BSC and other types of lung cancers had similarities in terms of age. But *Moro-Sibilot et al. (2008)* reported that BSC patients are older than non-BSC patients. Thus, *Wang et al. (2011)* demonstrated that there was no significant statistical difference of mean age between BSC (58.6 years) and poorly differentiated squamous cell carcinoma (60.5 years) ($P = 0.363$).There were more patients with poorly differentiated BSCs, while the numbers of patients of $N^+$ and $M^+$ were less than those with SCC, LCC and LAC. In our study, the 5-year survival rate of the BSC patients was close to 17.6%. In other reports, the 5-year survival rate for BSC of stage I and stage II was less than 15%, much lower than the 5-year survival rate of 47% for resectable poorly differentiated SCC (*Moro et al., 1994*). However, *Kim et al. (2003)* reported that there was no significant difference in the median survival rate between BSC and SCC in patients with stage I, without lymph node metastasis. Moreover, *Moro-Sibilot et al. (2008)* reported that operative modes had no difference between the prognosis of BSC and poorly-differentiated SCC. As shown in the Fig. S2, we compared the differences between the two groups by utilizing the survival curve. It clearly indicated that poorly-differentiated BSCs had better 5-year prognosis than poorly-differentiated SCCs, which were similar to the overall comparison results of this research. *Wang et al. (2011)* also revealed that BSC and poorly differentiated squamous cell carcinomas had very similar clinical features, and there were no significant differences in survival rates, while in our results the survival of poorly differentiated BSC was superior to that of SCC with the same differentiation. More research should be carried out to validate the results.

Currently, surgery is the best curative treatment in stage I, stage II, and some stage III non-small lung cancers (*Lang-Lazdunski, 2013*). Thus, lobectomy is still recommended as a preferred treatment for BSC, while more patients with peripheral tumors have undergone

sublobar section (*Zhang & Shen-Tu, 2015*). However, both in our univariate analyses and multivariate analyses, patients with lobectomy had a better prognosis than patients undergoing other therapies. Our results also suggested that at any stage, even stage III and IV, the prognosis of patients with lobectomy was significantly better than those with sublobar section. This may due to the radical lobectomy that reduces the potential risks for relapse and distant metastases of solid tumors (*Wang & Zhao, 2016*). In addition, survival following sublobar section was inferior to lobectomy for stage I non-small cell lung cancer (*Zhang et al., 2015b*). Therefore, further studies with larger cohorts, between lobectomy and sublobar section, especially when classified by histology, should be performed.

Nomogram, as an easily available and measurable tool of statistical prediction, which provides prognostic probability of specific outcomes (*Kent et al., 2016*; *Zhang et al., 2015a*). So far, multiple nomograms have been constructed for predicting prognosis of different types of lung cancers (*Zhang et al., 2017*; *Young et al., 2017*; *Ye et al., 2018*). Thus, it has even been considered more applied than the traditional AJCC TNM staging system in diverse malignancies according to great quantity of previous evidence (*Liang et al., 2015*; *Xie et al., 2015*). Furthermore, nomograms are especially advisable to deal with individual patients without existing definite clinical guidelines. In general, it seems simple and convenient via utilizing nomograms to predict patients' long-time survival according to their own characteristic.

The latest National Comprehensive Cancer Network recommends that *EGFR* mutations and other gene mutations should be considered as markers for lung squamous cell carcinoma, especially for non-smokers, small biopsy, or mixed squamous cell carcinoma (*Keedy et al., 2011*; *Felip et al., 2011*). Although the gene mutation status has not been well investigated in BSC, a molecularly targeted treatment may still have great potential to be used in the treatment for BSC.

The SEER database is a population-based tumor epidemiology database in the United States, covering about 28% of the population, including thousands of cases of lung cancers since 1973, therefore the SEER database is of great help in the study of lung cancer and other tumors (*Yang et al., 2017*; *Yang et al., 2018*). By analyzing the cases in the entire population of the SEER database, it is possible to effectively avoid the bias of the patients from the research given by a single institution. Nevertheless, there is often a lack of imaging data, smoking history, gene mutations, tumor markers, and data regarding other detailed treatments, especially chemotherapy regimens in the SEER database. Therefore, the impact of these factors on the prognoses of patients with BSC was not included in our study. These factors may significantly affect the prognoses of the patients.

In our study, we have selected BSC cases that met the requirements as much as possible. But there was still a significant gap with the number of SCC. Though there seemed to be some controversy, it was still determined by its specific characteristics. We should further pay close attention to the future prognosis of BSCs. We acknowledge that the article limited the findings to epidemiological analysis and did not set more emphasis on exploring the biology of rare tumors such as molecular mechanism for gene therapy strategy.

## CONCLUSION

BSC has unique clinical and prognostic features that differ from SCC, LCC and LAC. Age, T stage, N stage, M stage and surgery were found to be independent predictors of prognoses in patients with BSC. The nomogram we constructed better predicted the patients' 1-, 3-, and 5-year survival probabilities.

### Funding

This work was supported by the Suzhou Industry Technology Innovation Program (SYSD2017172), and the Suzhou Science and Technology Development Program (SYSD2018135). The funders had no role in study design, data collection and analysis, decision to publish, or preparation of the manuscript.

### Grant Disclosures

The following grant information was disclosed by the authors:
Suzhou Industry Technology Innovation Program: SYSD2017172.
Suzhou Science and Technology Development Program: SYSD2018135.

### Competing Interests

The authors declare there are no competing interests.

### Author Contributions

- Guangda Yuan, Cheng Zhan and Yiwei Huang conceived and designed the experiments, performed the experiments, analyzed the data, contributed reagents/materials/analysis tools, prepared figures and/or tables, authored or reviewed drafts of the paper.
- Donglin Zhu and Hongya Xie performed the experiments.
- Tengteng Wei and Tao Lu analyzed the data.
- Qun Wang conceived and designed the experiments, approved the final draft.
- Yong Yang and Yimeng Zhu conceived and designed the experiments, performed the experiments, analyzed the data, contributed reagents/materials/analysis tools, authored or reviewed drafts of the paper, approved the final draft.

### Data Availability

 The raw data is available in the Supplemental Files.

### Supplemental Information

Supplemental information for this article can be found online at http://dx.doi.org/10.7717/peerj.6724#supplemental-information.

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
