# Peer review of "Clinical characteristics and prognosis of basaloid squamous cell carcinoma of the lung: a population-based analysis"

_PeerJ, doi:10.7717/peerj.6724_

## Round 0.1 · original submission · Minor Revisions

Dear Dr. Zhu and Yang,

Our reviewers have gone through your manuscript and provided their opinion. Your manuscript can be accepted for publication only after the concerns raised by them are properly addressed. Make sure you address the concerns of Reviewer 1 about the lack of in depth literature review, few grammatical errors and the comment 1. Reviewer 2 has several suggestions that also need a thorough rebuttal. Before re-submitting it is highly recommended that you check for any grammatical and spelling errors that might have left been uncorrected (Eg: Line 41, 49, 106).

Reviewer 1 ·

Basic reporting

The manuscript entitled “Clinical characteristics and prognosis of basaloid
squamous cell carcinoma of the lung: a populationbased analysis” has analyzed the clinical features and prognosis of basaloid squamous cell carcinoma of the lung and constructed a nomogram to predict the prognoses of patients. However, it poorly quotes literature and lacks explanation on several fronts.

Experimental design

Fine

Validity of the findings

Fine

Additional comments

This study needs a major revision. My specific comments are:
1) In literature basaloid SCC has been shown to have a poor survival compared to Non-basaloid SCC. However, Authors conclusion is just opposite? (Reference: DOI:10.1158/1078-0432.CCR-14-0459 Published November 2014, DOI: 10.1183/09031936.00058507). Please explain in detail in light with the above quoted literature.
2) There is a huge difference in patient number between SCC (around 89000) and Basaloid SCC (425) which could affect survival curve to get biased.
3) Are the patient classified under Basaloid SCC, pure or mixed type?
4) Nomogram has not been explained in detail to emphasize its importance. More introduction and relevance should be added in the manuscript to conclude data.
5) “Worldwide in 2018, it accounted for 2.1 new cases and 1.8 million deaths”: Is it 2.1 million new cases?

Reviewer 2 ·

Basic reporting

The figures and tables are clearly described.
References include the most that are relevant. The authors do not miss any major referencing.
The article is clearly written.

Experimental design

• The classification of BSC as a subtype of non-small cell lung carcinomas (NSCC) is currently accepted by WHO organization but there is uncertainty on the nature of BSCs. Currently, BSCs are described to overlap features of both squamous cell carcinoma (SCC) and large cell carcinoma (LCC). Yet, it is not clear why authors focus their comparative studies to SCCs only. For a comprehensive analysis, BSS analysis must also be compared with LCC data.
• BSC is now regarded as a subtype of NSCC. In addition to the individual comparisons of BSCs with SCC and LCC, BSCs must also be compared to a cumulative set of NSCCs. It is essential to understand the prognosis and correlative features of BSCs within the NSCC category.
• Moro-Sibilot D et al (2008) reported that, BSC patients are older than non-BSC patients. The results here do not report a difference in median age. Is this a reflection of higher number of patients enrolled in this analysis compared to the previous?
• Include the difference in the prognosis on a subset of these tumors: 274 poorly-differentiated BSCs v/s 32,307 poorly-differentiated SCCs described in Table 1. This might help to address the discrepancies discussed in line 137 under the discussion section.

Validity of the findings

The manuscript summarizes the epidemiological analysis of clinical parameters on a rare set of basaloid squamous cell carcinoma of the lung (BSC) and describes the prognosis and correlations. The authors have used a previously available database, conduct the well-described statistical analyses and propose that, BSC have worst prognosis, display distinct correlations with tumor stages and suggest that their observations are clinically useful. The presented data is of high quality but the author’s interpretations are generalized and limits the findings of this manuscript to its epidemiological analyses and does not explore the biology of the rare tumors.

Additional comments

None.

---

## Round 0.2 · accepted · Accept

Congratulations! The reviewers have gone through your rebuttal and the revised manuscript and are satisfied with the changes incorporated and additional data and explanations provided.

Reviewer 1 ·

Basic reporting

Fine

Experimental design

Good

Validity of the findings

Fine

Additional comments

Authors have incorporated all the suggested changes. Manuscript can be accepted.

Reviewer 2 ·

Basic reporting

The revised manuscript has fixed the previous typos. The new manuscript is easy to read and the figures and tables are well organized.

Experimental design

No comment

Validity of the findings

The prognosis findings and clinical correlations is novel but the pathological analysis of the tumors is limited.

Additional comments

The authors have attempted to address the reviewer’s concerns and included additional figures, tables and text as requested. Additional data and explanations offered by the authors addresses the clinal prognosis and correlations of the basaloid squamous cell carcinoma of the lung (BSC) and strengthens the original observations presented before. The authors have also attempted to correlate their findings to the previously published articles and offered explanations where they observed an inconsistent finding. Due to the limited sample availability, the authors do not add new data to describe the pathological characteristics of BSC tumors, but the epidemiological analysis alone may of be interest to the researchers and clinicians.